# Whole genome sequence and comparative genome analyses of multi-resistant *Staphylococcus warneri* GD01 isolated from a diseased pig in China

**Canying Liu, Xianjie Zhao, Honglin Xie, Xi Zhang, Kangjian Li, Chunquan Ma, Qiang Fu**[ID] *

Department of Life Science and Engineering, Foshan University, Guangdong, China

* fuqiang@fosu.edu.cn

**Data Availability Statement:** All relevant data are within the paper and its Supporting Information files.

## Abstract

*Staphylococcus warneri* is a coagulase-negative staphylococcus that is a normal inhabitant of the skin. It is also considered to be an opportunistic etiological agent causing significant infections in human and animals. Currently, relatively little attention has been paid to the genome biology of *S. warneri* pathogenicity and antibiotic resistance, which are emerging issues for this etiological agent with considerably clinical significance. In this study, we determined the complete genome sequence of *S. warneri* strain GD01 recovered from the sampled muscle abscess tissue of a diseased pig in South China. The genome of *S. warneri* is composed of a circular chromosome of 2,473,911 base pairs as well as eight plasmid sequences. Genome-wide metabolic reconstruction revealed 82 intact functional modules driving the catabolism of respiration and fermentation for energy production, uptake of distinct sugars as well as two-component regulatory systems. The evidence uncovered herein enables better understanding for metabolic potential and physiological traits of this etiological agent. The antibiotic susceptibility test demonstrated that *S. warneri* GD01 was resistant to penicillin, amoxicillin, ampicillin, cefalexin, vancomycin, and sulfisoxazole. The associations between antibiotic phenotypes and the related genotypes were identified to reveal the molecular basis conferring resistance to this pathogen. A number of genes coding for potential virulence factors were firstly depicted in the genome of *S. warneri* GD01, including adhesins, exoenzymes, capsule, and iron acquisition proteins. Our study provides a valuable genomic context of the genes/modules devoting to metabolism, antibiotic resistance, and virulence of *S. warneri*.

## Introduction

Gram-positive bacterium *Staphylococcus warneri* is a common commensal as part of the normal flora colonizing human and animals' skin and mucosal membranes [1]. It is a facultatively anaerobic, nonmotile, coagulase-negative *staphylococcus* (CNS) species within the order of *Bacillales* [2]. Over the past three decades, clinical reports have pointed out that *S. warneri* is

**Funding:** This work was supported by the National Natural Science Foundation of China (31872443), Project of Department of Education of Guangdong Province (2017KTSCX193, 2014KTSPT037), and Key Laboratory for Preventive Research of Emerging Animal Diseases in Foshan University (KLPREAD201801-03).

**Competing interests:** The authors have declared that no competing interests exist.

an opportunistic etiological agent frequently isolated from the immunocompromised cases bearing bacteremia, sepsis with multiple abscesses, orthopedic infections, vertebral osteomyelitis, and ventricular shunt infections [3–7]. *S. warneri* has been also suggested to be a culprit of bovine abortion, indicating it is a zoonotic pathogen [8].

Like the other pathogenic staphylococci, the virulence of *S. warneri* have been suggested to be multifactorial, including adhesins, exoenzymes, capsule, iron uptake systems, and virulence regulators [2, 9]. A recent study on the pathogenesis of *S. warneri* infections has revealed the isolates from blood specimens are capable of adhesion to epithelial cells and forming biofilm with many expressed antibiotic resistance genes inside [9]. The molecular analyses via PCR has further indicated that biofilm formation of *S. warneri* is associated with the *icaADBC* genes as well as the other novel genes [9]. In addition, the patterns of multi-drug resistance (MDR) have been studied in 26 *S. warneri* strains derived from orthopedic infections, most cases associated with implant materials [5]. However, the genetic repertoire that contributes to the emergence of MDR and biosynthesis of virulence factors still needs comprehensive understanding at the whole-genome scale for this opportunistic pathogen.

The first complete genome sequence of *S. warneri* strain SG1 isolated from a laboratory is reported in 2013, Canada [10]. To date, the full genomes of five *S. warneri* strains are publicly available in NCBI Genome database (up to March, 2019). At the present study, we sequenced the complete genome of *S. warneri* GD01 isolated from muscle abscess tissue of a diseased pig in South China. Genome-wide metabolic analyses revealed the presence of genes/modules that plays a role in bacterial physiological and biochemical abilities. Through comparisons of the genic components between *S. warneri* GD01 and the other representative strains, we further focused on identifying the genes associated with phenotypic antibiotic resistance and bacterial virulence.

## Materials and methods

### Bacterial strain

In this study, the tissue used for isolating *S. warneri* strain GD01 was sampled from abdominal muscles of a pig from a commercial farm in March 2017 in South China. All experimental protocols were approved by the Animal Care and Use Committee of Guangdong Province and were performed accordingly. The approval ID or permit numbers were SCXK (Guangdong) 2015–0108. Through the serial dilution method, bacterial colonies were incubated on nutritional agar (Oxoid, United Kingdom) at 37°C for 24 h. After microscopic examination and 16S rRNA gene sequencing, this bacterial isolate was identified as staphylococci. The isolate was incubated in Luria-Bertani (LB) medium overnight at 37°C and the harvested cultures were stored at -40°C for further DNA extraction. This isolate was also subject to transmission electron microscopy (TEM) for bacterial morphology observation.

### Phenotypic characterization of AMR profiles

Based on the disk diffusion method described by BSAC guidelines [11], the antibiotic susceptibility of the isolate was assayed by the following antibiotics: penicillin (10 μg), ampicillin (10 μg), amoxicillin (10 μg), cefalexin (30 μg), cephadantine (30 μg), cefoxitin (30 μg), ceftriaxone (30 μg), streptomycin (10 μg), neomycin (30 μg), amikacin (30 μg), kanamycin (30 μg), gentamicin (10 μg), amikacin (30 μg), vancomycin (30 μg), tetracycline (30 μg), doxycycline (30 μg), sulfisoxazole (300 μg), sulfamethoxazole (25 μg), norfloxacin (10 μg), ofloxacin (5 μg), ciprofloxacin (5 μg), rifampin (5 μg).

## 16S rRNA gene sequencing

Genomic DNA was extracted using a Blood & Cell Culture DNA Mini Kit (Qiagen, Hilden, Germany). The harvested DNA was used as template for the PCR analyses of the 16S rRNA gene using universal bacterial 16S rRNA primers forward-P (5′-AGAGTTTGATCCTGGCTCAG-3′) and reverse-P (5′-ACGGCTACCTTGTTACGACTT-3′), which could amplify approximately 1500 bp fragment [12]. The PCR reaction was conducted at 98˚C for 3 mins followed by 25 cycles of 98˚C for 30 secs, 56˚C for 30 secs and 72˚C for 90 secs, and 72˚C for 5 mins. The positive product was purified and then sequenced by using an ABI 3730 DNA sequencer (Applied Biosystems, CA, USA). Using the NCBI BLAST server (https://blast.ncbi.nlm.nih.gov/Blast.cgi), the nearly full-length sequence of 16S rRNA was searched against Prokaryotic 16S ribosomal RNA database.

## Whole genome sequencing and assembly

A whole genome shotgun strategy was employed and sequencing experiments were performed on both platforms of Illumina HiSeq and Pacific Biosciences RS II, respectively. A Library for Illumina sequencing was prepared using the TruSeqTM DNA Sample Prep Kit (Illumina Inc., CA, USA) following the manufacturer's recommendation. Genomic DNA (gDNA) was quantified by the Qubit dsDNA BR Assay kit (Life Invitrogen) and was sheared into ~300–500 bp fragments using Covaris M220 instrument (Covaris, MA, USA). A paired-end library with ~350-bp insertion fragments was then constructed for bridge PCR amplification using TruSeq PE Cluster Kit v3-cBot-HS (Illumina Inc., CA, USA). Sequencing reactions were carried out on a HiSeq using Truseq SBS Kit v3-HS. In total, 10,071,010 paired-end reads were produced. Using Trimmomatic v0.36 [13], raw reads with the adaptor sequences and low quality tails were trimmed and filtered according to the following criteria: average quality score of 20; the tailed bases with minimum quality score of 20; read length of 50 bp. After quality control, 9,642,261 high-accuracy short reads were retained for correcting the long-read sequences. For PacBio sequencing, gDNA was sheared into ~10 kb fragments using a Covaris G-tube (Covaris, MA, USA). After purification of fragmented gDNA, a SMRTbell library was then constructed using PacBio SMRTbell template Prep Kit 1 (Pacific Biosciences, CA, USA) according to the manufacturer's protocols. The resulting library was sequenced using P6-C4 chemistry on a PacBio RS II machine (Pacific Biosciences, CA, USA). The PacBio sequencing generated 36,843 long reads with a mean read length of 12.5 kb. According to the algorithm previously described [14], the long reads were subject to a hybrid error correction by mapping short reads to them and then assembled by Celera Assembler 8.0 [15]. The coverage was estimated by mapping the clean reads to the genome assemblies (2,544,623 bp in total), resulting in a ~550-fold coverage.

## Genome annotation and comparative genomics

Taxonomic inference of the newly sequenced genome was carried out using the method for calculating Average Nucleotide Identity (ANI) implemented by a Python module pyani (https://github.com/widdowquinn/pyani). For this application, complete genomes of six *S. warneri* strains and ten strains from five closely related *Staphylococcus* species were collected. Pairwise genome sequence alignments using BLASTN v 2.5.0+ [16] were performed for any paired genomes across all strains. ANI was subsequently calculated based on the aligned regions based on the algorithm described by Richter et al. [17]. Genome annotations of genetic elements were conducted by using the integrative analyses pipeline Prokka v1.13 [18]. Briefly, protein-coding sequences (CDSs), transfer RNAs (tRNAs) and ribosomal RNAs (rRNAs) were predicted using Prodigal v2.6.3 [19], Aragorn v1.2.38 [20] and Barrnap v0.9 [18], respectively.

**Table 1. Genome features of the *S. warneri* GD01 and five *S. warneri* representative strains.**

| Strain | GD01 | 22.1 | SWO | NCTC7291 | NCTC11044 | SG1 |
|---|---|---|---|---|---|---|
| Chromosome | | | | | | |
| GenBank accession No. | CP038242 | CP032159 | CP033098 | LR134244 | LR134269 | CP003668 |
| Sequence length (bp) | 2,473,911 | 2,515,743 | 2,466,231 | 2,451,975 | 2,427,576 | 2,486,042 |
| GC content (%) | 32.8 | 32.9 | 32.7 | 32.8 | 32.8 | 32.7 |
| CDSs | 2,349 | 2,407 | 2,361 | 2,324 | 2,322 | 2,359 |
| 16S rRNA | 6 | 6 | 5 | 6 | 6 | 5 |
| 23S rRNA | 6 | 6 | 5 | 6 | 6 | 5 |
| 5S rRNA | 7 | 7 | 6 | 7 | 7 | 6 |
| Transfer RNAs | 62 | 62 | 62 | 62 | 62 | 59 |
| Plasmids | | | | | | |
| Sequence length (bp) | P1:32338 | P1:25886 | P1:53165 | None | None | P1:19866 |
| | P2:8656 | | P2:30898 | | | P2:16515 |
| | P3:7760 | | P3:22366 | | | P3:13186 |
| | P4:5392 | | | | | P4:8232 |
| | P5:4697 | | | | | P5:6212 |
| | P6:4525 | | | | | P6:4374 |
| | P7:4439 | | | | | P7:3352 |
| | P8:2905 | | | | | P8:2937 |

The prophage element was predicted using the PHASTER web server [21]. The predicted CDSs were functionally annotated using a curated database from UniProtKB [18] and protein functional categorizing was performed using the COG (Clusters of Orthologous Groups of proteins) database [22]. For the bacterial proteome, KEGG metabolic pathways and functional modules were analyzed using BLASTKOALA [23]. To get a glimpse of chromosomal structure and genome-wide sequence conservation, pairwise genome alignments between GD01 as query and the other *S. warneri* strains as subject were conducted using BLASTN v 2.5.0+ [16]. A circular map of genome signatures was produced using the CGView Comparison Tool server [24]. To estimate bacterial pangenome structure, Roary v3.12 [25] was employed to cluster orthologous genes present in the six fully sequenced genomes of *S. warneri* strains GD01, 22.1, SWO, NCTC7291, NCTC11044, and SG1 (The corresponding GenBank accession numbers are shown in Table 1). A maximum-likelihood phylogenetic tree was reconstructed based on the core-genome SNP alignments produced by the Parsnp v1.2 package [26].

## Bioinformatics analyses of resistome and virulome

To predict potential antimicrobial resistance (AMR) genes, the BLASTP (version 2.3.0+) [16] analysis was performed against the Comprehensive Antibiotic Resistance Database (CARD) v3.0.1 [27]. The BLASTP output was further parsed to identify the AMR genes according to the following parameters: E-value cutoff of 1e-20, minimum alignment identity of 40%, and the subject coverage of at least 60%. Bacterial virulence-associated genes were detected by using BLASTP to search against the experimentally verified candidates collected by the Virulence Factor Database (VFDB) [28]. The top hit with E-value cutoff of 1e-20 was retained and the corresponding query was categorized by virulence factors. Global alignment for certain pairs of homologues was conducted using the Needleman-Wunsch algorithm implemented by the NCBI BLAST server. Hmmer v3.2.1 [29] was used to search protein functional domains based on the Pfam-A database v32.0 [30]. The Pfam protein family domain was extracted if the best hit satisfying the E-value cutoff of 1e-04. Bacterial protein subcellular localization was

predicted using the PSORTb v3.0.2 server with the default options for Gram-positive bacteria [31].

### Nucleotide sequence accession numbers

The complete genome sequences of the chromosome and plasmids of *S. warneri* strain GD01 have been submitted to the GenBank database under the BioProject PRJNA512551 with the accession numbers CP038242-50.

## Results and discussion

In this study, the bacterial strain (designated as GD01 hereafter) was recovered from the sampled muscle abscess tissue of a diseased pig. To infer taxonomic assignment of GD01, we initially performed PCR amplification and Sanger sequencing of the 16S rRNA gene fragment. Based on the BLASTN analyses, the sequenced 16S rRNA fragment (1475 bp; GenBank No.: MG214350) was found to share 99% identity with 16S rRNA (1470 bp; NR_025922) of *S. warneri* strain AW25. Transmission electron microscopy (Fig 1A) showed the morphological features of GD01 were typical for cocci-like bacteria in the genus of *Staphylococcus* [2]. To further accurately determine the taxonomy and genetic diversity of GD01, whole genome sequencing was carried out by a combination strategy of short-read and long-read sequencing. Genomic characterizations and comparative genomic analyses were depicted and discussed below in details.

### General features of the genome

Genome assemblies of strain GD01 are composed of a complete circular chromosome (2,473,911 bps) and eight plasmids (70,712 bps in total). Whole genome level ANI measures were obtained to infer taxonomic assignment of GD01. Fig 1B displays a heatmap of the ANI values between pairs of full genomes from 16 *Staphylococcus* strains including GD01. The newly sequenced GD01 genome shares 99.27~99.84% ANI values with those of five *S. warneri* strains 22.1, NCTC7291, NCTC11044, SG1, and SWO, respectively (S1 Table). Since 95% ANI has been considered as a typical percentage threshold for the species boundary [17], strain GD01 should belong to *S. warneri*. Additionally, it was obvious that all the tested strains from different species in the genus of *Staphylococcus* were clustered into six blocks according to their taxonomic affiliations. Notably, the genome of *S. warneri* GD01 shares 83.50% and 83.63% ANI with the two genomes of *S. pasteuri* strains 3C and JS7, respectively. Furthermore, relatively lower ANI values with a range from 77.25% to 78.21% were observed between the genomes of GD01 and the other four *Staphylococcus* species (S1 Table). ANI-based analyses again confirmed that *S. warneri* was most closely related to *S. pasteuri* among the species within the genus *Staphylococcus* [32].

Statistics of genomic annotations of GD01 are summarized and compared with the other representative *S. warneri* strains (Table 1). The overall G+C content (GC%) of the *S. warneri* GD01 genome is 32.8%, which is similar to those of the other *S. warneri* genomes with GC% ranged from 32.7% to 32.9%. The chromosome of *S. warneri* GD01 encodes 2,349 protein-coding sequences (CDSs), 62 tRNA genes and 19 rRNA genes. Of these CDSs, 1,961 (~83.5%) were assigned to 21 general COG functional categories (Table 2). Except for the chromosomally encoded genic components, totally 81 CDSs were predicted in the plasmid sequences belonging to GD01. Obviously, genes coding for products involved in the COG category "Mobilome: prophages, transposons" were over-represented in the genetic repertoire of the plasmids (~11.1%) compared to the chromosome backbone (~0.7%) of GD01. In addition, two intact prophage regions designated Phage_1 (43.4 kb, 59 CDSs, G+C content of 33.8%)

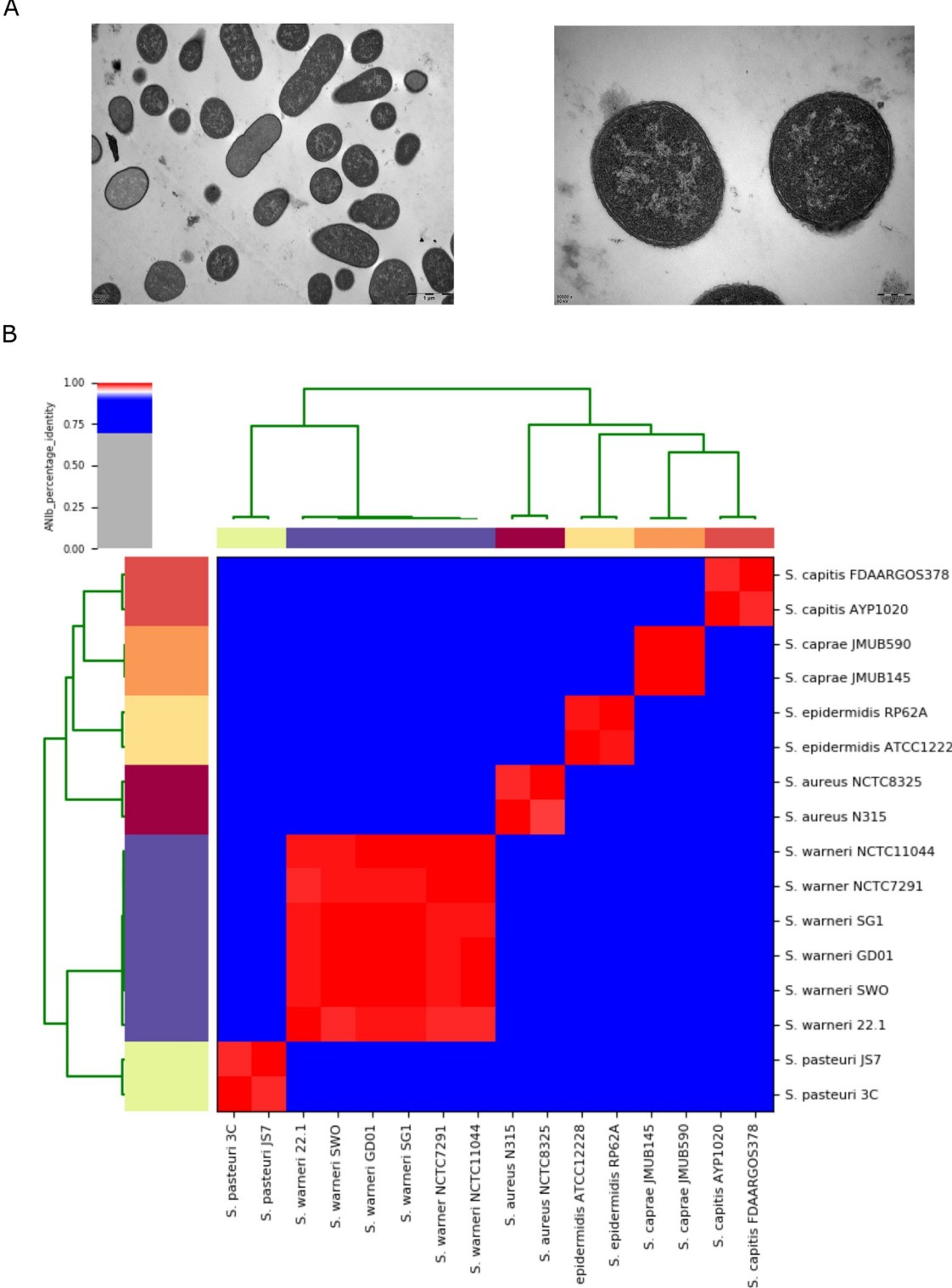

**Fig 1. Morphological and taxonomic characteristics of the clinical isolated strain GD01.** A). Transmission electron micrograph of GD01. Bacterial sections (60 nm) were stained with 2% uranyl acetate and lead citrate and then viewed by the Hitachi H-7600 TEM (Hitachi, Japan). B). Heatmap of the ANI values between pairs of genomes from 16 strains belonging to six *Staphylococcus* species. The ANI values are color-coded by the top left color bar. The GenBank accession numbers of the complete genome sequences analyzed herein as well as the ANI values are listed in S1 Table.

**Table 2. COG functional classification for the protein-coding genes on the chromosome and plasmids of *S. warneri* GD01.**

| COG ID | COG category | Chromosome | | Plasmids | |
|---|---|---|---|---|---|
| | | No. | Percent (%) | No. | Percent (%) |
| - | Not in COGs | 388 | 16.5 | 43 | 53.1 |
| C | Energy production and conversion | 91 | 3.9 | 0 | 0 |
| D | Cell cycle control, cell division, chromosome partitioning | 23 | 1 | 4 | 4.9 |
| E | Amino acid transport and metabolism | 158 | 6.7 | 0 | 0 |
| F | Nucleotide transport and metabolism | 63 | 2.7 | 0 | 0 |
| G | Carbohydrate transport and metabolism | 120 | 5.1 | 4 | 4.9 |
| H | Coenzyme transport and metabolism | 90 | 3.8 | 0 | 0 |
| I | Lipid transport and metabolism | 60 | 2.6 | 0 | 0 |
| J | Translation, ribosomal structure and biogenesis | 188 | 8 | 0 | 0 |
| K | Transcription | 107 | 4.6 | 4 | 4.9 |
| L | Replication, recombination and repair | 94 | 4 | 4 | 4.9 |
| M | Cell wall/membrane/envelope biogenesis | 105 | 4.5 | 2 | 2.5 |
| N | Cell motility | 2 | 0.1 | 1 | 1.2 |
| O | Posttranslational modification, protein turnover, chaperones | 74 | 3.2 | 0 | 0 |
| P | Inorganic ion transport and metabolism | 105 | 4.5 | 4 | 4.9 |
| Q | Secondary metabolites biosynthesis, transport and catabolism | 24 | 1 | 0 | 0 |
| R | General function prediction only | 140 | 6 | 0 | 0 |
| S | Function unknown | 167 | 7.1 | 1 | 1.2 |
| T | Signal transduction mechanisms | 48 | 2 | 0 | 0 |
| U | Intracellular trafficking, secretion, and vesicular transport | 19 | 0.8 | 1 | 1.2 |
| V | Defense mechanisms | 49 | 2.1 | 2 | 2.5 |
| X | Mobilome: prophages, transposons | 16 | 0.7 | 9 | 11.1 |
| MCC | Multiple COG Categories | 218 | 9.3 | 2 | 2.5 |

and Phage_2 (14.8 kb, 23 CDSs, G+C content of 30.7%) were detected in the *S. warneri* GD01genome (Fig 2). Interestingly, about two thirds (n = 14) of the genes present in the Phage_2 are homologous to those encoded by the phage PT1028 isolated from a mitomycin C-treated culture of *S. aureus* NY940 (NC_007045 15.6 kb). Pairwise nucleotide sequence alignment showed that Phage_2 shares 52% identity with PT1028, indicating both phages may be derived from a common ancestry.

Orthologous gene clustering on the total predicted CDSs from six *S. warneri* genomes resulted in a pangenome containing 3,212 genes (S2 Table). Of these, 65% (n = 2,103) were found to be core genes that were possessed by all *S. warneri* strains, 10% (n = 309) were accessary genes present in at least two strains but not all, and the remaining (n = 800) were strain-specific genes. Each strain contains 135 unique genes in average. The presence/absence of certain genes associated with antibiotic resistance and virulence across strains was discussed below detaily.

## Genome-scale metabolic potential of *S. warneri*

To get a glimpse of bacterial metabolic potential, the predicted proteome of *S. warneri* was analyzed with the database of KEGG metabolic pathway and functional module. As a result, 1,405 CDSs of *S. warneri* GD01 were assigned to 166 KEGG functional modules, accounting for ~57.8% of the GD01 proteome. Nearly half (n = 82) of these functional modules were found to be complete, which enabled producing metabolites and protein complexes to implement the known physiological and biochemical characterizations for this staphylococcal species (S3

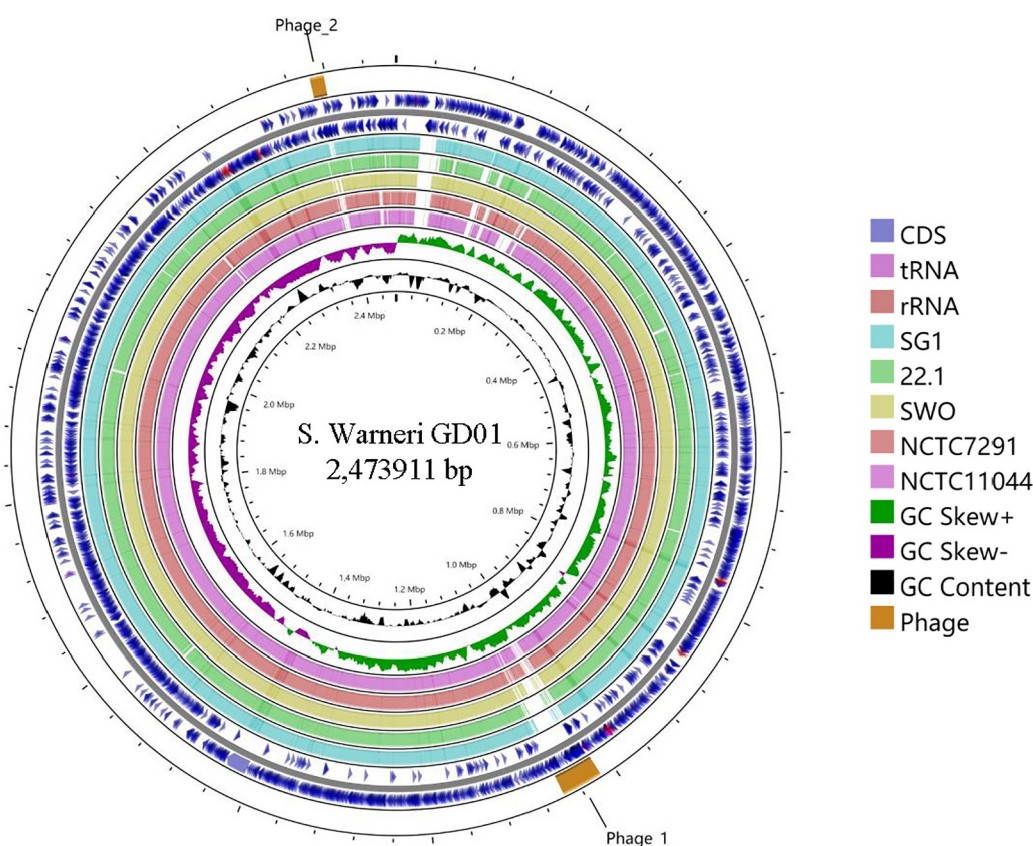

**Fig 2. Circular representation of the complete genome of *S. warneri* GD01 and the comparison with other *S. warneri* strains.** Circles are indexed from the outside to the inside. The outermost two circles represent protein-coding genes (blue), tRNAs (pink) and rRNAs (red) genes on the forward and reverse strands of the GD01 genome. Using BLASTN matches (E-value cutoff of $10^{-10}$), circles 3 to 7 denote pairwise genomic sequence conservation between five *S. warneri* strains (SG1, 22.1, SWO, NCTC7291, NCTC11044) and the newly sequenced strain GD01. The innermost two circles show the plots of GC skew and mean centered GC content of the GD01 genome. The prophage element is displayed with the outside arc in orange.

Table). For instance, a series of intact modules associated with ATP production through respiration and fermentation were identified in the genome of GD01, including the tricarboxylic acid (TCA) cycle (M00009-11), oxidative and non-oxidative pentose phosphate pathway (M00006-7), glycolysis and gluconeogenesis (M00001-M00003), and pyruvate oxidation (M00307). The Entner-Doudoroff pathway as an alternative involved in glucose metabolism was absent in *S. warneri*, due to the gene *edd* encoding phosphogluconate dehydratase was missing in the GD01 genome. The patterns of these metabolic units again support a previous option that *S. warneri* is facultatively anaerobic microorganism [2].

A number of modules that comprise genes encoding phosphotransferase system responsible for the uptake of different sugars were identified in the GD01 genome, e.g. glucose (M00809), trehalose (M00270), fructose (M00273), mannitol (M00274), and lactose (M00281). Additionally, the functional modules with complete sets of genes devoting to carbon fixation (M00579), dissimilatory nitrate reduction (M00530), and assimilatory sulfate reduction (M00176), were also identified in the genome of *S. warneri* GD01.

Bacterial regulatory machines, especially two-component systems (TCSs), harbor the capability to alter cellular metabolism and mediate adaptive responses in various niches [33]. Eight pairs of TCSs encoded by *phoRB* (M00434), *resED* (M00458), *vicKR* (M00459), *bceSR*

(M00469), *desKR* (M00479), *liaSR* (M00481), *nreBC* (M00483), and *agrCA* (M00495), were identified in the genome of *S. warneri* GD01 (S3 Table). Moreover, GD01 possesses an unannotated TCS pair encoding a response regulator ArlR and a signal transduction histidine kinase ArlS, which are homologous to the *S. aureus* ArlRS that enable the pathogen to overcome calprotectin-induced manganese starvation [34]. Additionally, *S. warneri* is urease positive bacterium [2]. The operon *ureABCEFGD* (0628–0622) encoding urease structural proteins UreABC and accessory proteins UreEFGD is present in the GD01 genome, which is homologous to the *S. aureus* urease gene cluster whose products are required for a persistent murine kidney infection [35]. *S. warneri* GD01 also possesses complete sets of genes coding for protein complexes, which constitute the classical Sec and Tat secretion systems (M00335-336).

## Antibiotic resistance analyses

Based on BLASTP searching against the CARD database, we identified 19 genes associated with antimicrobial resistance (AMR) in the genome of *S. warneri* GD01. The majority of the AMR genes are localized on the chromosome except for the tetracycline resistance gene *tet(K)* carried by the plasmid p7GD01. Details of these AMR genes and their resistant potential for antibiotics are summarized in Table 3. These gene products are divided into two groups, antibiotic efflux pump proteins (i.e. *ykkD*, *ykkC*, *norB1*, *norB2*, *arlR2*, *arlS*, *tetA*, *mgrA*, *mepR*, *mepA*, *tet(K)*) and antibiotic targets (i.e. *gyrB*, *parE*, *folA*, *walR*, *phoP*, *uppP*, *rpoB*, *sul4*). Bacterial efflux pumps are generally transport proteins localized in the cytoplasmic membrane. Eight of 11 efflux pump related genes detected herein encode the proteins located in the cytoplasmic membrane (Table 3).

The disk susceptibility test demonstrated that *S. warneri* GD01 was resistant to the following six antibiotics: penicillin, amoxicillin, ampicillin, cefalexin, vancomycin, and sulfisoxazole. The AMR genes associated with these antibiotic phenotypes were particularly focused on below. The gene *mgrA* of *S. warneri* GD01 encodes an HTH-type transcriptional regulator (147 aa), which shares 92.5% identity with 99% coverage of a *Staphylococcus* homologue (WP_001283444, 147 aa). MgrA belong to both AMR gene families of major facilitator superfamily (MFS) antibiotic efflux pump and ATP-binding cassette (ABC) antibiotic efflux pump, which are responsible for directed pumping of antibiotic out of a cell to confer resistance [36]. MgrA also possesses a MarR family domain (PF01047; 1.4e-14) and a winged helix DNA-binding domain (PF13463; 5.9e-08) which could regulate the expression of the *mar* operon involved in the extrusion of multiple antibiotics from within cells into the external niches [37]. The MDR caused by MgrA relates to many classes of antibiotics, including beta-lactam, fluoroquinolone, peptide antibiotics, and so on. Consistently, *S. warneri* GD01 was resistant to penicillin, amoxicillin, ampicillin, and cefalexin, all of which belong to beta-lactam antibiotics.

In *S. warneri* GD01, two transcriptional regulators encoded by the genes *walR* (233 aa) and *phoP* (236 aa) were detected, both of which belong to the AMR gene family of glycopeptide resistance and may confer resistance to vancomycin. In addition, WalR of *S. warneri* shares 46% identity with a transcriptional activator VanRI (232 aa) found in *Desulfitobacterium hafniense*, which has been reported to confer resistance to vancomycin in many Gram-positive bacterial species [38]. Notably, WalR and PhoP are both encoding a response regulator receiver domain (PF00072) responsible for receiving the signals from the sensor partner of two-component systems. In addition, the *folP* gene (267 aa) of *S. warneri* GD01 encodes a dihydropteroate synthase that shares 59% similarity with 94% coverage of a homologue coding for a sulfonamide resistant protein Sul4 (WP_102607457, 287 aa) prevalent in a large metagenomic dataset, whose mobilization across distinct microbial communities has led to widespread of sulfonamide resistance in humans and animals [39]. The presence of *folP* in GD01

**Table 3. Genes encoding proteins with a potential role in antibiotic resistance of *S. warneri* strain GD01.**

| Symbol | Product function | Protein accession (Identity %) | AMR Gene Family | Class of drug | Localization[a] |
|---|---|---|---|---|---|
| *gyrB* | DNA gyrase subunit B | AAO47226 (44) | aminocoumarin resistant parY; aminocoumarin self resistant parY | aminocoumarin antibiotic | CP |
| *walR* | Transcriptional regulatory protein | WP_011461303 (46) | glycopeptide resistance gene cluster; vanR | glycopeptide antibiotic | CP |
| *norB1* | Quinolone resistance protein | CCQ22388 (52) | MFS antibiotic efflux pump | fluoroquinolone antibiotic | CM |
| *norB2* | Quinolone resistance protein | CCQ22388 (54) | MFS antibiotic efflux pump | fluoroquinolone antibiotic | CM |
| *mepR* | hypothetical protein | YP_001440920 (48) | multidrug and toxic compound extrusion (MATE) transporter | glycylcycline; tetracycline antibiotic | CP |
| *mepA* | Multidrug export protein | AAU95768 (80) | MATE transporter | glycylcycline; tetracycline antibiotic | CM |
| *ykkD* | Multidrug resistance protein | CAB13167 (53) | small multidrug resistance (SMR) antibiotic efflux pump | aminoglycoside antibiotic; tetracycline antibiotic; phenicol antibiotic | CM |
| *ykkC* | Multidrug resistance protein | CAB13166 (55) | SMR antibiotic efflux pump | aminoglycoside antibiotic; tetracycline antibiotic; phenicol antibiotic | CM |
| *phoP* | Alkaline phosphatase synthesis transcriptional regulatory protein | AEP40503 (42) | glycopeptide resistance gene cluster; vanR | glycopeptide antibiotic | CP |
| *folA* | Dihydrofolate reductase | AAO04716 (83) | trimethoprim resistant dihydrofolate reductase dfr | diaminopyrimidine antibiotic | CP |
| *arlR* | Response regulator | WP_000192137 (85) | MFS antibiotic efflux pump | fluoroquinolone antibiotic; acridine dye | CP |
| *arlS* | Signal transduction histidine-protein kinase | YP_499945 (69) | MFS antibiotic efflux pump | fluoroquinolone antibiotic; acridine dye | CM |
| *parE* | DNA topoisomerase 4 subunit B | AAO47226 (42) | aminocoumarin resistant parY; aminocoumarin self resistant parY | aminocoumarin antibiotic | CP |
| *tetA* | Tetracycline resistance protein, class B | AAS68233 (85) | MFS antibiotic efflux pump | fluoroquinolone antibiotic; acridine dye | CM |
| *mgrA* | HTH-type transcriptional regulator | WP_001283444 (93) | ABC antibiotic efflux pump; MFS antibiotic efflux pump | fluoroquinolone antibiotic; cephalosporin; penam; tetracycline antibiotic; peptide antibiotic; acridine dye | CP |
| *uppP* | Undecaprenyl-diphosphatase | AAC76093 (46) | undecaprenyl pyrophosphate related proteins | peptide antibiotic | CM |
| *rpoB* | DNA-directed RNA polymerase subunit beta | BAD59497 (62) | rifamycin-resistant beta-subunit of RNA polymerase | peptide antibiotic; rifamycin antibiotic | CP |
| *folP* | Dihydropteroate synthase | WP_102607457 (41) | sulfonamide resistant sul | sulfonamide antibiotic | CP |
| *tet(K)* | transport system protein | YP_003283625 (100) | MFS antibiotic efflux pump | tetracycline antibiotic | CM |

[a] The abbreviations of protein subcellular localization are CP for cytoplasmic and CM for cytoplasmic membrane.

may devote to bacterial resistance to sulfisoxazole, one of the sulfonamide antibiotics. However, more experimental works and incoming sequenced genomes are needed to uncover the chromosomal point mutations that enable antibiotic resistance in *S. warneri*.

## Gene patterns of virulence factors in *S. warneri*

Clinical infection caused by *S. warneri* has been progressively reported since 1984 [4]. However, the genetic basis of the virulence factors (VFs) for this emerging pathogen still lacks comprehensive investigation. According to sequence similarity searching against the VFDB database, 201 genes encoding putative virulence-associated products were identified in the pan-genome of *S. warneri*. The majority (88%) of these genes were highly conserved and present in all six *S. warneri* genomes. Among these genes, 183 are present in the GD01 genome (S4

Table). The presence/absence patterns of the virulence-associated genes assigned to six categories (i.e. capsule, immune evasion, adherence, exoenzyme, iron uptake, and secretion system) across all strains are shown in Fig 3. Additionally, according to the prediction of subcellular localization, the number of genes encoding cytoplasmic membrane protein, cell-wall protein, and extracellular protein is 75, 1, and 9, respectively. These cell-surface associated virulence genes may contribute special roles on bacterial adherence, biofilm formation, antiphagocytosis, and immune evasion for this opportunistic pathogen.

Bacterial adherence is often initiative for subsequent biofilm formation of gram-positive *Staphylococci* [9]. Genes associated with adhesion and biofilm were identified in the *S. warneri* GD01 genome (Fig 3). The *ebpS* gene of *S. warneri* GD01 encodes an elastin-binding protein (608 aa) that is a predicted cell-surface protein sharing 52% identity with Ebp (486 aa) of *S. aureus* MW2, which could promote bacterial colonization to facilitate pathogenesis [40]. Gene *yloA* (568 aa) coding for a fibronectin-binding protein shares 85% identity with an Ebh homologue (WP_049400214, 565 aa) of *S. epidermidis*, which may be involved in biofilm formation and bacterial adherence to both biotic and abiotic surfaces [41]. The gene *sdrC* encodes Ser-Asp repeat-containing protein C (1,483 aa), which comprises a typical SdrD B-like domain (PF17210), followed by Ser-Asp dipeptide repeats and a LPXTG anchor motif (PF00746) at the C-terminus. The characteristics of such structural organizations are uniformly found in *Staphylococci* SdrC proteins [42]. Notably, the genes *icaADBC* encoding polysaccharide intercellular adhesion, which is a main component for *S. aureus* biofilm [9], are absent in the genome of *S. warneri* GD01.

Capsular polysaccharides (CPSs) are diverse structural components that lie outside the cell envelope of Gram-positive bacteria. As an important virulence factor, CPSs are often expressed by many pathogens to escape from host immune responses, e.g. *Streptococcus* and *Staphylococcus* species [43]. Approximately 21 genes encoding products involved in the synthesis of *S. warneri* capsule were detected in the genome of GD01 (S4 Table). For instance, the genes *ywqE* and *mnaA* encoding products are homologous to the CPS biosynthetic enzymes from *S. aureus*. Two genes *cpsD_1* (590 aa) and *cpsD_2* (467 aa) of *S. warneri* GD01 code for products belonging to the same glycosyltransferase family 2 (PF00535) responsible for adding sugar monomers to make the glycans. Except for capsule, it seems that *S. warneri* is likely to produce adenosine, which has been studied its crucial roles on antiphagocytosis [44]. For this attribute, *S. warneri* GD01 possesses a gene *mggB* coding for adenosine synthase A (EOJ31_2166, 887 aa), which shares 59% identity with *S. aureus* AdsA (KDP49196, 967 aa) that is a cell wall-anchored enzyme involved in *staphylococci* virulence and abscess formation as well as host immune evasion.

Bacterial exoenzymes are secreted extracellular components, which are considered to be a class of virulence factors in pathogenic *staphylococci* usually encompassing proteases and lipases [45]. In the GD01 genome, ten genes code for putative exoenzymes predicted to be localized at the extracellular space (Fig 3). The gene *sspA* encode serine protease (316 aa) that is homologous to the *S. aureus* V8 protease. An extracellular elastase encoded by *sepA_1* (506 aa) shares 57% aa identity with zinc metalloproteinase aureolysin Aur (509 aa) of *S. aureus*. In addition, two genes *sspP* (EOJ31_2143, 388 aa) and *sspB* (EOJ31_02250, 396 aa) code for staphopain A and B, respectively, the homologs of which are the major secreted cysteine proteases of *S. aureus*. The extracellular proteases mentioned above have been reported to be crucial factors mediating virulence of *S. aureus* [46], probably suggesting similar mechanisms in pathogenesis of *S. warneri*. Furthermore, both genes (*lip_1*, 733 aa; *lip_2*, 746 aa) annotated as triacylglycerol lipase precursor were found to be a pair of paralogues, sharing 63% aa sequence similarity with each other. Intriguingly, the third lipase gene *lip2* (EOJ31_2397, 688 aa) present in the GD01 genome shares 99% aa sequence identity with a *S. epidermidis* homologue

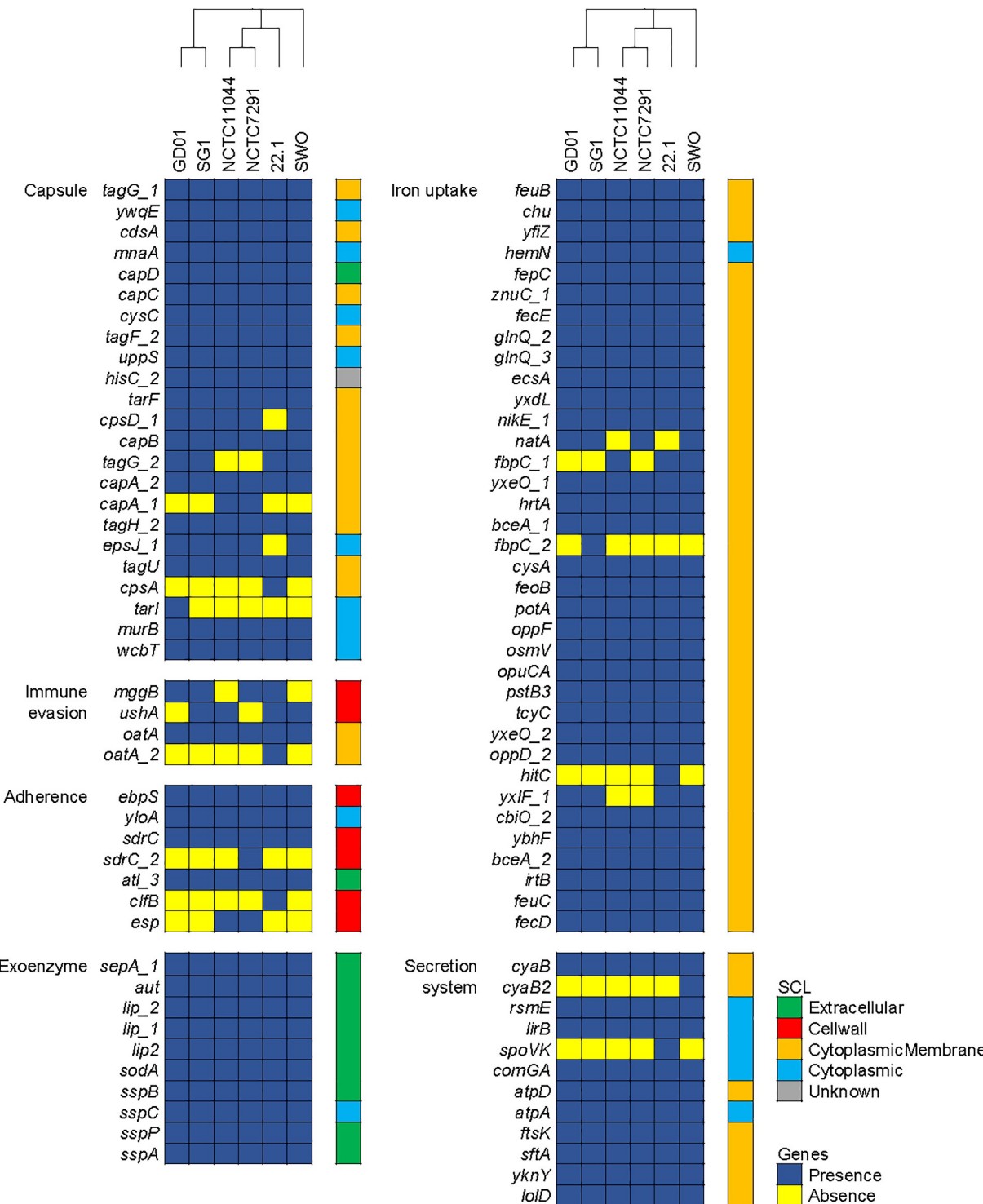

**Fig 3. Genetic distribution and protein subcellular localization of the genes encoding putative virulence factors in *S. warneri*.** The upper panel is corresponding to the core SNP phylogeny. The presence and absence of virulence-associated genes are categorized and color coded in blue and yellow, respectively. The classification of protein subcellular localization (SCL) for each gene is color-coded as following: extracellular in green, cell wall in red, orange in cytoplasmic membrane, cytoplasmic in cyan, and unknown in grey.

(WP_126510627, 688 aa), implicating its horizontal transfer across staphylococcal species. All these lipase genes comprise a YSIRK type signal peptide at the N-terminus (PF04650).

As well known, iron is important nutrition for bacterial survival and growth. A large number of VF genes associated with iron acquisition and utilization were present in the genome of *S. warneri*, including FbpABC, enterobactin, heme permease protein Chu, and ferrous iron transporter FeoB (Fig 3). Interestingly, according to the subcellular localization, nearly all of these iron metabolism related proteins were located at the cytoplasmic membrane, constituting iron transport complexes involved in the uptake of extracellular iron.

## Conclusions

In summary, we sequenced and obtained a complete genome of an MDR *S. warneri* GD01 isolated from swine in China. Genome-wide metabolic reconstruction revealed a full spectrum of compact functional modules and protein complexes driving the catabolism of respiration and fermentation for energy production, uptake of distinct sugars as well as two-component regulatory systems. The evidence uncovered herein enables better understanding for metabolic potential and physiological traits of this etiological agent. The associations between antibiotic phenotypes and the related genotypes were detected to reveal the putative molecular mechanism conferring resistance to penicillin derivatives, cephalosporins, and vancomycin in *S. warneri* GD01. This study sheds light on genomic context of the genes/modules devoting to metabolism, antibiotic resistance, and virulence of *S. warneri*.

## Supporting information

**S1 Table. The ANI values between pairs of genomes among 16 strains from different Staphylococcus species.**
(XLSX)

**S2 Table. Presence and absence of orthologous genes across S. warneri genomes.**
(XLSX)

**S3 Table. The list of KEGG functional modules identified in the genome of *S. warneri* GD01.**
(XLSX)

**S4 Table. The list of genes encoding putative virulence factors in *S. warneri*.**
(XLSX)

## Acknowledgments

We are grateful for excellent technological support of Zhuofei Xu and Kai Wang at Shanghai MasScience Biotechnology Co., Ltd.

## Author Contributions

**Conceptualization:** Canying Liu.

**Data curation:** Canying Liu.

**Formal analysis:** Canying Liu.

**Funding acquisition:** Qiang Fu.

**Investigation:** Canying Liu, Xianjie Zhao, Honglin Xie, Xi Zhang, Kangjian Li, Chunquan Ma.

**Methodology:** Canying Liu, Xianjie Zhao, Honglin Xie, Xi Zhang, Kangjian Li, Chunquan Ma.

**Project administration:** Qiang Fu.

**Resources:** Qiang Fu.

**Software:** Canying Liu.

**Supervision:** Canying Liu, Qiang Fu.

**Validation:** Qiang Fu.

**Visualization:** Qiang Fu.

**Writing – original draft:** Canying Liu.

**Writing – review & editing:** Canying Liu, Qiang Fu.

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
