## [Decision Letter · Decision Letter 0]

3 Oct 2019

PONE-D-19-22604

Whole genome sequence and comparative genome analyses of multi-resistant Staphylococcus warneri GD01 isolated from a diseased pig in China

PLOS ONE

Dear Mr Fu,

Thank you for submitting your manuscript to PLOS ONE. After careful consideration, we feel that it has merit but does not fully meet PLOS ONE’s publication criteria as it currently stands. Therefore, we invite you to submit a revised version of the manuscript that addresses the points raised during the review process.

We would appreciate receiving your revised manuscript by Nov 17 2019 11:59PM. To enhance the reproducibility of your results, we recommend that if applicable you deposit your laboratory protocols in protocols.io, where a protocol can be assigned its own identifier (DOI) such that it can be cited independently in the future. For instructions see: http://journals.plos.org/plosone/s/submission-guidelines#loc-laboratory-protocols

We look forward to receiving your revised manuscript.

Kind regards,

Keun Seok Seo, Ph.D.

Academic Editor

PLOS ONE

Journal Requirements:

2. We understand that you obtained tissue from a diseased pigs for this study. In your Methods section, please provide additional information regarding the source of the animal:(1) name of the company, farm, or store, (2)where the tissue was obtained from the pig and (3) the cause of death.

Additional Editor Comments (if provided):

Reviewers' comments:

Reviewer's Responses to Questions

**Comments to the Author**

1. Is the manuscript technically sound, and do the data support the conclusions?

Reviewer #1: Yes

Reviewer #2: Yes

2. Has the statistical analysis been performed appropriately and rigorously? 

Reviewer #1: N/A

Reviewer #2: Yes

3. Have the authors made all data underlying the findings in their manuscript fully available?

Reviewer #1: Yes

Reviewer #2: Yes

4. Is the manuscript presented in an intelligible fashion and written in standard English?

Reviewer #1: Yes

Reviewer #2: Yes

5. Review Comments to the Author

Reviewer #1: The authors present a complete S. warneri genome sequence, completed using a combination of PacBio and Illumina HiSeq sequencing. The strain is phenotypically and genotypically characterised, and confirmation of species achieved using Average Nucleotide Identity (ANI). While the work has been performed to a high standard, some small adjustments could be made, to make methods/results clearer, and in terms of the ANI analysis to strengthen the conclusion drawn.

Major Comments

Line 42: “The other part was used for isolation of bacterial colonies”. On what agar? How was the tissue handled/plated? “Certain colonies”. How were these selected?

Line 60: “which amplify approximately 1500 bp fragment [12]. Using the NCBI BLAST server”. The authors jump from PCR amplification straight to analysis of the sequence. How was this amplicon sequenced?

Line 80: What chemistry was used for PacBio sequencing?

Lines 145-148: The authors compare the ANI values of S. warneri against S. epidermidis and S. aureus. But S. warneri is most closely related to S. pasteuri, while S. caprae and S. capitis are also in the same clade as S. warneri/S. pasteuri/S. epidermidis (see phylogeny from Grana-Miraglia et al. 2018 PeerJ). The inclusion of genomes from S. pasteuri, S. caprae and S. capitis in the ANI analysis would be more meaningful. Complete genomes for all three are available via Genbank.

Minor Comments

Line 58: 16S not 16s

Line 74: “Using Trimmomatic”. Version number?

Line 88-90: “Taxonomic assignment…. implemented by OrthoANI Tool v0.93.1”. What reference genomes were used for this?

Similarly on line 100: “pairwise genome alignments were conducted using BLASTN”. Against what? What references were being BLASTed against?

Similarly line 102-103: “Roary v3.12 was employed to cluster orthologous genes and define bacterial pangenome”. Again, what other strains were used to define the pangenome?

Line 129: “sequencing of the partial 16S”. Sanger sequencing?

Line 168: should this be “six S. warneri”, rather than “six S. aureus”?

Line 169: “Of these, 65% were found”. Would be useful to give the actual number of genes here (e.g. Of these, 65% (n= ) were found).

Line 266-7: “The presence/absence patterns of these virulence-associated genes across all strains are shown in Figure 3”. Figure 3 only shows 92 genes , not the 183 or 201 mentioned earlier in the paragraph. How were these 92 selected?

Line 301: (2166, 887 aa). Is 2166 the locus number? It almost looks like the protein is 2,166,887aa. A locus tag, in front of the number, would be useful here.

Line 385: “the positive and negative strains of the GD01 genome”. What does this mean?

Reviewer #2: This manuscript provides a succinct description of a complete genome sequence of an opportunistic pathogen S. warneri, of importance to understanding a major source of morbidity and mortality on the planet (S. aureus). They appear to have properly sequenced (Pac Bio + illumina) and assembled the organism and eight plasmids, which is unusual for emerging pathogen samples. The presentation is logical and appropriate. I would have appreciated a more thorough analysis of the plasmids (where did they likely come from?) and am a little surprised there were not more AMR genes on them, but this isn't my paper. Similarly, the grammar would benefit from a careful proofreading (eg. 3rd sentence of the abstract reads "..genome biology of S. warneri has been paid to less attention on the pathogenicity and antibiotic resistance...", instead of something like, "relatively little attention as been paid to the genome biology of S. warneri pathogenicity and antibiotic resistance").

6. PLOS authors have the option to publish the peer review history of their article (what does this mean?). If published, this will include your full peer review and any attached files.

Reviewer #1: No

Reviewer #2: No

---

## [Author Response · Author response to Decision Letter 0]

31 Mar 2020

We would like to thank the helpful comments and suggestions from the associate Editor and the two reviewers, which have significantly improved our manuscript. Below are our point by point response to the comments.

Journal Requirements:

— We have revised the manuscript according to PLOS ONE's style requirements.

2. We understand that you obtained tissue from a diseased pigs for this study. In your Methods section, please provide additional information regarding the source of the animal:(1) name of the company, farm, or store, (2)where the tissue was obtained from the pig and (3) the cause of death.

—We have rewritten this paragraph in the revised manuscript as following “In this study, the tissue used for isolating S. warneri strain GD01 was sampled from abdominal muscles of a pig from a commercial farm in March 2017 in South China.”.

Reviewers' comments:

Reviewer #1: 

General comments:

The authors present a complete S. warneri genome sequence, completed using a combination of PacBio and Illumina HiSeq sequencing. The strain is phenotypically and genotypically characterised, and confirmation of species achieved using Average Nucleotide Identity (ANI). While the work has been performed to a high standard, some small adjustments could be made, to make methods/results clearer, and in terms of the ANI analysis to strengthen the conclusion drawn.

— We would like to thank you very much for your comments and suggestions.

Major Comments

Line 42: “The other part was used for isolation of bacterial colonies”. On what agar? How was the tissue handled/plated? “Certain colonies”. How were these selected?

— We have revised this paragraph as following “Through the serial dilution method, bacterial colonies were incubated on nutritional agar (Oxoid, United Kingdom) at 37℃ for 24 h. After microscopic examination and 16S rRNA gene sequencing, this bacterial isolate was identified as staphylococci. The isolate was incubated in Luria-Bertani (LB) medium overnight at 37℃ and the harvested cultures were stored at -40℃ for further DNA extraction. This isolate was also subject to transmission electron microscopy (TEM) for bacterial morphology observation.”.

Line 60: “which amplify approximately 1500 bp fragment [12]. Using the NCBI BLAST server”. The authors jump from PCR amplification straight to analysis of the sequence. How was this amplicon sequenced?

— The details on PCR amplification and sequencing were added in the revised manuscript as following “The PCR reaction was conducted at 98℃ for 3 mins followed by 25 cycles of 98℃ for 30 secs, 56℃ for 30 secs and 72℃ for 90 secs, and 72℃ for 5 mins. The positive product was purified and then sequenced by using an ABI 3730 DNA sequencer (Applied Biosystems, CA, USA).”.

Line 80: What chemistry was used for PacBio sequencing?

— The PacBio sequencing chemistry is P6-C4. This paragraph was revised as following “For PacBio sequencing, gDNA was sheared into ~10 kb fragments using a Covaris G-tube (Covaris, MA, USA). After purification of fragmented gDNA, a SMRTbell library was then constructed using PacBio SMRTbell template Prep Kit 1 (Pacific Biosciences, CA, USA) according to the manufacturer’s protocols. The resulting library was sequenced using P6-C4 chemistry on a PacBio RS II machine.”.

Lines 145-148: The authors compare the ANI values of S. warneri against S. epidermidis and S. aureus. But S. warneri is most closely related to S. pasteuri, while S. caprae and S. capitis are also in the same clade as S. warneri/S. pasteuri/S. epidermidis (see phylogeny from Grana-Miraglia et al. 2018 PeerJ). The inclusion of genomes from S. pasteuri, S. caprae and S. capitis in the ANI analysis would be more meaningful. Complete genomes for all three are available via Genbank.

—We agree with your opinion. We have downloaded complete genomes from S. pasteuri, S. caprae, and S. capitis. ANI-based analyses for pairwise genome comparison was performed using 16 strains belonging to six Staphylococcus species. We have rewritten the related methodology and results. We have revised the heatmap for visualizing ANI values between pairs of genomes.

—The methodology has been rewritten as following “Taxonomic inference of the newly sequenced genome was carried out using the method for calculating Average Nucleotide Identity (ANI) implemented by a Python module pyani (https://github.com/widdowquinn/pyani). For this application, complete genomes of six S. warneri strains and ten strains from five closely related Staphylococcus species were collected. Pairwise genome sequence alignments using BLASTN v 2.5.0+ [16] were performed for any paired genomes across all strains. ANI was subsequently calculated based on the aligned regions based on the algorithm described by Richter et al. [17].”.

—The result has been rewritten as following “Whole genome level ANI measures were obtained to infer taxonomic assignment of GD01. Figure 1B displays a heatmap of the ANI values between pairs of full genomes from 16 Staphylococcus strains including GD01. The newly sequenced GD01 genome shares 99.27~99.84% ANI values with those of five S. warneri strains 22.1, NCTC7291, NCTC11044, SG1, and SWO, respectively (Table S1). Since 95% ANI has been considered as a typical percentage threshold for the species boundary [17], strain GD01 should belong to S. warneri. Additionally, it was obvious that all the tested strains from different species in the genus of Staphylococcus were clustered into six blocks according to their taxonomic affiliations. Notably, the genome of S. warneri GD01 shares 83.50% and 83.63% ANI with the two genomes of S. pasteuri strains 3C and JS7, respectively. Furthermore, relatively lower ANI values with a range from 77.25% to 78.21% were observed between the genomes of GD01 and the other four Staphylococcus species (Table S1). ANI-based analyses again confirmed that S. warneri was most closely related to S. pasteuri among the species within the genus Staphylococcus [32].”.

Minor Comments

Line 58: 16S not 16s

— It has been revised according to your suggestion.

Line 74: “Using Trimmomatic”. Version number?

—It has been revised to “Using Trimmomatic v0.36”.

Line 88-90: “Taxonomic assignment…. implemented by OrthoANI Tool v0.93.1”. What reference genomes were used for this?

—We have added some genomes from the other Staphylococcus species according to your suggestion. For ANI analysis and visualization, a Python module pyani (https://github.com/widdowquinn/pyani) was used in the revised manuscript. In the ANI analysis, the genome of strain 1 is aligned against that of strain 2 for all pairs of genomes tested. Any genome could be used as reference once in pairwise genome comparison. We have rewritten the related methodology as following “Taxonomic inference of the newly sequenced genome was carried out using the method for calculating Average Nucleotide Identity (ANI) implemented by a Python module pyani (https://github.com/widdowquinn/pyani). For this application, complete genomes of six S. warneri strains and ten strains from five closely related Staphylococcus species were collected. Pairwise genome sequence alignments using BLASTN v 2.5.0+ [16] were performed for any paired genomes across all strains. ANI was subsequently calculated based on the aligned regions based on the algorithm described by Richter et al. [17].”.

Similarly on line 100: “pairwise genome alignments were conducted using BLASTN”. Against what? What references were being BLASTed against?

— The complete genome sequence of GD01 was used as query to search against five S. warneri strains as subject respectively using BLASTN. We have revised the sentence as following “pairwise genome alignments between GD01 as query and the other S. warneri strains as subject were conducted using BLASTN v 2.5.0+ [16].”.

Similarly line 102-103: “Roary v3.12 was employed to cluster orthologous genes and define bacterial pangenome”. Again, what other strains were used to define the pangenome?

— This sentence has been revised as following “To estimate bacterial pangenome structure, Roary v3.12 [25] was employed to cluster orthologous genes present in the six fully sequenced genomes of S. warneri strains GD01, 22.1, SWO, NCTC7291, NCTC11044, and SG1 (The corresponding GenBank accession numbers are shown in Table 1).”.

Line 129: “sequencing of the partial 16S”. Sanger sequencing?

— Yes, Sanger sequencing was used for sequencing 16S rRNA gene fragment. This sentence has been revised as following “To infer taxonomic assignment of GD01, we initially performed PCR amplification and Sanger sequencing of the 16S rRNA gene fragment.”.

Line 168: should this be “six S. warneri”, rather than “six S. aureus”?

— Yes, we have corrected the wrong description herein. The revised sentence is as following “Orthologous gene clustering on the total predicted CDSs from six S. warneri genomes resulted in a pangenome containing 3,212 genes (Table S1).”. A list shown in Table S1 has been added to display the presence/absence of orthologous genes across six S. warneri genomes.

Line 169: “Of these, 65% were found”. Would be useful to give the actual number of genes here (e.g. Of these, 65% (n= ) were found).

— We have revised this sentence as following “Of these, 65% (n = 2,103) were found to be core genes that were possessed by all S. warneri strains, 10% (n = 309) were accessary genes present in at least two strains but not all, and the remaining (n = 800) were strain-specific genes.”.

Line 266-7: “The presence/absence patterns of these virulence-associated genes across all strains are shown in Figure 3”. Figure 3 only shows 92 genes , not the 183 or 201 mentioned earlier in the paragraph. How were these 92 selected?

—We have revised the improper description herein. This sentence is revised as following “The presence/absence patterns of the virulence-associated genes assigned to six categories (i.e. capsule, immune evasion, adherence, exoenzyme, iron uptake, and secretion system) across all strains are shown in Figure 3.”. Many virulence factors were initially detected by using blastp searching against the VFDB database, and certain VF terms in the VFDB may contribute the similar pathogenic function, like “Capsule” and “Capsule I”, both of which are involved in the biosynthesis of bacterial capsule. For this limitation, we manually assigned the genes to some major categories and focus on six virulence factor categories according to the literatures and curated information on the Major virulence factors in Staphylococcus (http://www.mgc.ac.cn/cgi-bin/VFs/genus.cgi?Genus=Staphylococcus).

Line 301: (2166, 887 aa). Is 2166 the locus number? It almost looks like the protein is 2,166,887aa. A locus tag, in front of the number, would be useful here.

— It has been revised according to your suggestion as following “… S. warneri GD01 possesses a gene mggB coding for adenosine synthase A (EOJ31_2166, 887 aa)…”.

Line 385: “the positive and negative strains of the GD01 genome”. What does this mean?

— We have corrected the wrong spelling herein. The revised sentence is as following “The outermost two circles represent protein-coding genes (blue), tRNAs (pink) and rRNAs (red) genes on the forward and reverse strands of the GD01 genome.”.

Reviewer #2: 

General comments:

This manuscript provides a succinct description of a complete genome sequence of an opportunistic pathogen S. warneri, of importance to understanding a major source of morbidity and mortality on the planet (S. aureus). They appear to have properly sequenced (Pac Bio + illumina) and assembled the organism and eight plasmids, which is unusual for emerging pathogen samples. The presentation is logical and appropriate. I would have appreciated a more thorough analysis of the plasmids (where did they likely come from?) and am a little surprised there were not more AMR genes on them, but this isn't my paper. Similarly, the grammar would benefit from a careful proofreading (eg. 3rd sentence of the abstract reads "..genome biology of S. warneri has been paid to less attention on the pathogenicity and antibiotic resistance...", instead of something like, "relatively little attention has been paid to the genome biology of S. warneri pathogenicity and antibiotic resistance").

— We would like to thank you very much for your comments and suggestions.

— For the plasmid analysis, we further performed a Pfam domain analysis to investigate the putative plasmid replication protein families across the plasmids in GD01. We detected seven genes were encoding plasmid replication proteins in six of all assembled circular plasmids, including P1_0006 (RepA_N, PF06970), P1_0031 (Rep_1, PF01446), P2_0007 (Rep_3, PF01051), P4_0001 (Rep_trans, PF02486), P6_0001 (Rep_trans, PF02486), P7_0004 (Rep_trans, PF02486), P8_0003 (Rep_1, PF01446). Since we found the associations between the plasmids/replication proteins and the plasmid-borne AMR genes is not very clear in S. warneri, we then focused on genetic and functional characterizations of the GD01 chromosome in this study. We will check the plasmid diversity and verify them using more field strains from different niches in the subsequent study.

— The 3rd sentence of the abstract has been revised according to your suggestion. The revised sentence is as following “Currently, relatively little attention has been paid to the genome biology of S. warneri pathogenicity and antibiotic resistance, which are emerging issues for this etiological agent with considerably clinical significance.”.

— We are sorry for our poor English. The revised MS has been improved.

---

## [Decision Letter · Decision Letter 1]

5 May 2020

Whole genome sequence and comparative genome analyses of multi-resistant Staphylococcus warneri GD01 isolated from a diseased pig in China

PONE-D-19-22604R1

Dear Dr. Fu,

We are pleased to inform you that your manuscript has been judged scientifically suitable for publication and will be formally accepted for publication once it complies with all outstanding technical requirements.

With kind regards,

Keun Seok Seo, Ph.D.

Academic Editor

PLOS ONE

Additional Editor Comments (optional):

Reviewers' comments:

Reviewer's Responses to Questions

**Comments to the Author**

1. If the authors have adequately addressed your comments raised in a previous round of review and you feel that this manuscript is now acceptable for publication, you may indicate that here to bypass the “Comments to the Author” section, enter your conflict of interest statement in the “Confidential to Editor” section, and submit your "Accept" recommendation.

Reviewer #1: All comments have been addressed

2. Is the manuscript technically sound, and do the data support the conclusions?

Reviewer #1: Yes

3. Has the statistical analysis been performed appropriately and rigorously? 

Reviewer #1: N/A

4. Have the authors made all data underlying the findings in their manuscript fully available?

Reviewer #1: Yes

5. Is the manuscript presented in an intelligible fashion and written in standard English?

Reviewer #1: Yes

6. Review Comments to the Author

Reviewer #1: The authors have addressed all previous comments. A few minor suggestions regarding grammar:

Fig 1B contains a spelling error in the species name for NCTC7291 (it should be warneri, not warner).

Figure 2 should have a lowercase 'w' for warneri.

Line 20: should be 'considerable' rather than 'considerably'

Line 48: 'has been suggested' rather than 'have been suggested'

Line 119: should this be 'minimum read length'?

Line 136: I don't think 'collected' is the right word here. Maybe 'included' or 'analysed'.

Line 215: 'in total' rather than 'totally'

Line 232: 'on average'

Line 233: 'in detail' rather than 'detailly'

7. PLOS authors have the option to publish the peer review history of their article (what does this mean?). If published, this will include your full peer review and any attached files.

Reviewer #1: No

---

## [Editor Report · Acceptance letter]

13 May 2020

PONE-D-19-22604R1 

Whole genome sequence and comparative genome analyses of multi-resistant Staphylococcus warneri GD01 isolated from a diseased pig in China 

Dear Dr. Fu:

I am pleased to inform you that your manuscript has been deemed suitable for publication in PLOS ONE. Congratulations! Your manuscript is now with our production department. 

With kind regards,

on behalf of

Dr. Keun Seok Seo 

Academic Editor

PLOS ONE